

# Skill ranking of researchers via hypergraph

Xiangjie Kong[1], Lei Liu[1], Shuo Yu[1], Andong Yang[1], Xiaomei Bai[2] and Bo Xu[1]

[1] Key Laboratory for Ubiquitous Network and Service Software of Liaoning Province, School of Software, Dalian University of Technology, Dalian, China
[2] Anshan Normal University, Computing Center, Anshan, China

## ABSTRACT

Researchers use various skills in their works, such as writing, data analysis and experiments design. These research skills have greatly influenced the quality of their research outputs, as well as their scientific impact. Although many indicators have been proposed to quantify the impact of researchers, studies of evaluating their scientific research skills are very rare. In this paper, we analyze the factors affecting researchers' skill ranking and propose a new model based on hypergraph theory to evaluate the scientific research skills. To validate our skill ranking model, we perform experiments on the PLOS ONE dataset and compare the rank of researchers' skills with their papers' citation counts and h-index. Finally, we analyze the patterns about how researchers' skill ranking increased over time. Our studies also show the change patterns of researchers between different skills.

## INTRODUCTION

The burst of development in science contributes to an expansion of knowledge and technology. It also leads to many new disciplines and interdisciplines emerging in universities (*Lee, Xia & Roos, 2017*). Scientific cross-disciplinary collaboration brings positive effects to improve the productivity and quality of researchers' outputs. Collaboration has become a common phenomenon in people's daily life for a long time. More and more works are finished with the form of collaboration, such as project and assignment, patent, software development and scientific papers. The patterns of collaboration and teamwork have attracted the interest of researchers in many disciplines (*Kong et al., 2017*). Organizations like funding agencies, universities, enterprises, etc. are also concerned about team-based issues to improve the productivity and boost the profits. Many state-of-art works have been proposed to analyze the pattern of collaboration and optimize the team structure. Many efforts have been made to analyze and optimize teams in terms of their topology structure (*Milojević, 2014*; *Kong et al., 2018*; *Wang et al., 2017*), which have made great contributions. However, team effectiveness not only depends on the appropriate team topology structure, but also depends on their function component, such as the abilities and skill distributions of team members (*Li et al., 2017*). Some works built, evaluated and refined teams in consideration of both skills of team members and the team topology structure (*Li et al., 2017*; *Wang, Zhao & Ng, 2016*).

Corresponding author
Xiaomei Bai,
xiaomeibai@outlook.com

Quantifying individual's ability and impact is a key point for solving the problems of team building, evaluation and refinement. It is also needed in many other practical applications; for example, awards granting, score assessment and scholarship awarding. However, how to evaluate a researcher's ability with limited resources in the diverse academic circumstance is still under exploration.

Under the scenario of big scholarly data, a large number of data-driven indices and measures were proposed to evaluate the impact of researchers reliably (*Xia et al., 2017*). In the early years, h-index (*Hirsch, 2005*) was proposed to evaluate both the publication quantity and quality of researchers based on the number of papers and the citations of the highly cited paper. A researcher with an index $h$ means that he/she has $h$ published papers and each of them has been cited at least $h$ times. The index has been proved to be a simple but valid measure in evaluating scholars' scientific impact (*Bornmann & Daniel, 2005*). The h-index has become a frequently used index and has been applied to solve many problems by scholars. Besides, many other indices are proposed to evaluate the performance and impact of researchers, such as g-index (*Egghe, 2006*), S-index (*Sawilowsky, 2012*) and AIF (*Pan & Fortunato, 2013*). Another frequently used measure is Q parameter (*Sinatra et al., 2016*), which is a unique parameter to capture the ability and evaluate the impact of a scientist. In recent works, methods of ranking authors in a heterogeneous network have been proposed (*Meng & Kennedy, 2013*; *Liu et al., 2014*), in which multi-types of vertices and relationships are taken into consideration. However, these measures are proposed to evaluate researchers' abilities and impact on the macro level. They cannot reflect a researcher's proficiency with a particular skill. Using these measures to solve some practical problems may influence the accuracy of the results.

Researchers' skill sets have been used in solving many problems, such as team formation, team member replacement and team refinement (*Kong et al., 2016*). Many researchers use terms extracted from the paper's keywords and title or conference and journal name as authors' skills (*Li et al., 2017*). They only consider a binary relationship between researchers and skills, that is, whether a researcher has a particular skill. However, in real-life practice, the relationship between researcher and skill is more complicated. The skillfulness of an expert should be taken into consideration according to his previous experience. *Farhadi et al. (2011)* proposed a skill grading method in team formation problem. They proposed a complex formula based on the similarity between scholars and their possessed skills to calculate the skillfulness level. Many works used authors' contribution information to measure their impact and analyzed collaboration patterns of authors (*Persson, 2017*; *Paul-Hus et al., 2017*; *Rahman et al., 2017*; *Corrêa Jr. et al., 2017*; *Biswal, 2013*). The contribution information can also be used in credit allocation and ranking authors (*Sekercioglu, 2008*; *Dance, 2012*). The contributions are mainly extracted from acknowledgements of papers, or contribution information in journals like PLOS journals, the British Medical Journal and the FEBS Journal. Other skill evaluation methods are proposed for measuring workers' skills, where the skills are extracted from job seekers' resumes in the online job market (*Zhou et al., 2017*; *Anderson, 2017*) and online social platform (*Alvarez-Rodríguez & Colomo-Palacios, 2014*). For example, in economic issues, skills are extracted for analyzing the relationship between skill and wage (*Anderson, 2017*)

and reducing the skills gap (*Zhou et al., 2017*). However, these methods are mainly proposed to solve problems in labor economics, but cannot be used to evaluate the skills for students and researchers. Despite their success in evaluating a researcher's impact and ability to some extent, a fine-grained method that quantifies researchers' skill level is still needed now. In this paper, we analyze the factors affecting scientific skill ranking, and construct a heterogeneous network for mapping them into a hypergraph. We present measures to calculate the weights of the edges in the heterogeneous network. The degree of skillfulness for a researcher is denoted as the weight of the hyperedge, which is calculated by a method inspired from the Laplace kernel function. Our contributions are summarized as follows:

- Model establishment. We carry out data and empirical analysis on the features that influence the proficiency of researchers' skills, and then we establish a Skill Ranking Model (SRModel) to evaluate the skillfulness of researchers via hypergraph.
- Dataset construction. We construct a dataset by crawling information from PLOS ONE journal website, including 164,543 papers authored by 684,844 researchers with 10 kinds of skills aggregated by contributions.
- Skill ranking analysis. We perform our model on the PLOS ONE journal dataset to validate the effectiveness of the model and the pattern of how scholars' skill ranking increased over time.

## BACKGROUND

Facing the highly competitive academic society, researchers have to master a wider variety of skills and knowledge to improve personal strength. Besides, the discipline integration and the researcher's skill migration have become a trend in academia with the rapid development of science and technology. It is a great challenge to rank skills of researchers under such complex conditions. In this part, we aim at figuring out what influence the researchers' skill ranking according to data analysis and empirical analysis. We use data collected from the PLOS ONE journal website to make an investigation and carry out our analysis. The PLOS ONE dataset contains over 170,000 articles collected from the journal website. These papers' publication years range from 2006 to June 2017. The dataset includes paper title, publish year, authors, number of citations, disciplines and more specifically, authors' contributions. More details will be introduced in the 'Experiment Setting' section.

Science and technology are playing important roles in promoting the growth of productivity owing to the unprecedented achievement the researchers have made recently. This development tendency also leads to a burst of complex tasks that need to be solved by working together among experts in different fields. The disciplinary collaboration has reduced the boundaries between disciplines and resulted in a sharp emergence of many marginal disciplines. It is clear that collaboration cross discipline becomes a trend and this will be more frequent in the future. Take PLOS ONE journal, for example; we extracted the disciplines of the PLOS ONE paper on their website, and the statistics of those 177,147 papers are shown in Fig. 1. There are only 1,843 papers in a single field, and a large number of papers cross 2–4 fields.

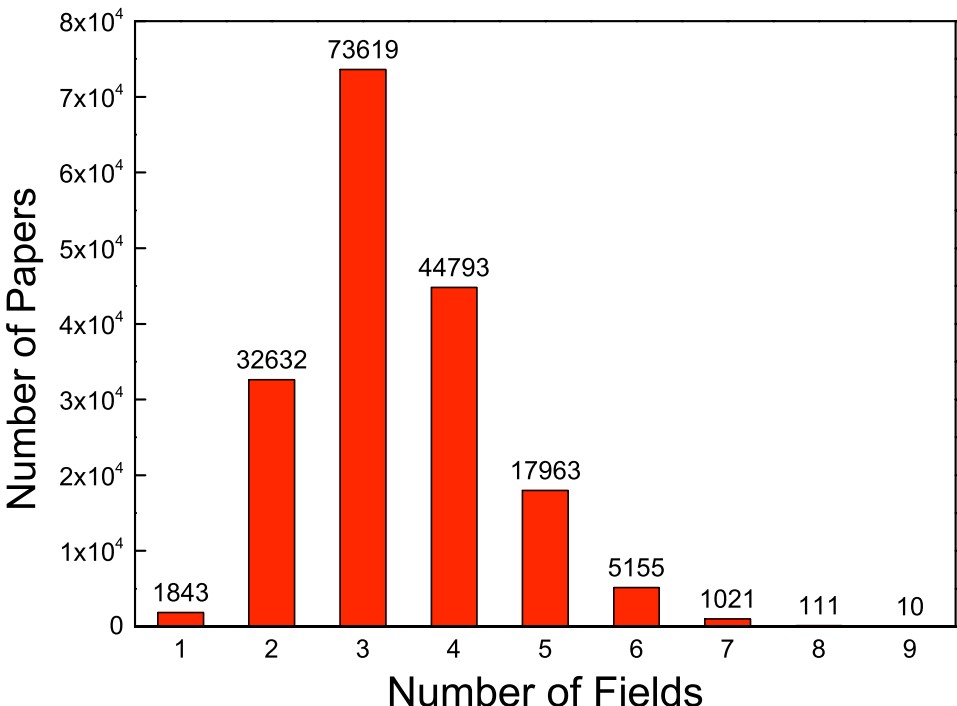

**Figure 1** The distribution of number of fields for all the papers.

It has been found that collaborations cross disciplines have brought many competitive and high-quality outputs (*Yegros-Yegros, Rafols & D'Este, 2015*; *Nicolini, Mengis & Swan, 2012*). First, the development of the collaborative relationships between multi-disciplines makes researchers realize the advantages of resource sharing, which can make a better data and knowledge utilization either within a team or in the whole society. The efficiency and outputs of the team are also improved under the cross-disciplinary scenario. Second, cross-disciplinary collaboration can bring creative ideas to researchers and produce more novel outputs for teams. Researchers from different fields get strong support from other disciplines during collaboration and discussion. They are more likely to discover valuable problems and create significant innovations because they have less limitation than those who collaborate with researchers in the same field.

As cross-disciplinary cooperation has become a trend in academia, discipline is one of the most important factors when considering team-related problems. Researchers can learn and use skills of several disciplines in a cross-disciplinary work, which involves knowledge of more than one discipline. However, scientific research varies in different fields, and skills required by each field are diverse. One kind of skill can be different in different fields or disciplines. For example, the skill "experiment" in computer science is totally different from that in biology and chemistry, while another skill "paper writing" is more similar between various disciplines or fields. Thus, the skill in different fields has both common features and uniqueness. The diversity of fields and disciplines should be taken into consideration while evaluating researchers' skills.

After taking discipline as an important factor in skill ranking problem, we have two questions: (i) in what aspects it may influence the skill ranking and (ii) how can we quantify a skill's ranking? Firstly, we consider the importance of a skill in a field, which indicates the relevance of skills in each field. As discussed above, the significance of the same skill in different fields varies, which is caused by discipline diversity and speciality. Here we suppose that the more frequently a skill is used in a field, the more important it is in that field. Thus, we use the rank of times that a skill has been used in a field to quantify its importance in that field.

Secondly, a researcher always masters multiple skills, and the proficiency of each skill is not equal as well. As a common sense, the more frequently people use a skill, the more proficient they are in it, that is "Skill comes from practice". This indicates that the times a researcher use a skill in works are correlated to the skill's ranking. Similarly, we use the rank of times that researchers use a given skill in their works to denote skill's importance to researchers.

Besides, the familiarity of researchers with a certain research field is also vital for ranking skills of researchers in different fields. This feature is quantified by the rank of number of papers that a researcher published in a given field. The more papers a researcher has published in a field, the more familiar he/she is with that field. Thus, they can perform better and show a higher skill proficiency more easily.

According to the above analysis, in order to rank researchers' skill, field information needs to be taken into consideration. Importance of a skill in a field, a researcher's proficiency of each skill, and the familiarity of a researcher in a field can influence the ranking level of a skill. Considering those factors, we propose a novel model to rank researchers' skill.

## MODEL

This section describes the definition of skill ranking and the concept of hypergraph. We describe our Skill Ranking Model (**SRModel**) based on the hypergraph in detail.

### Problem definition

We define the skill ranking problem as follow: given a complex network $H = (V, E, w)$, where $V$ denotes the vertices, including researchers, fields and skills in our model, $E$ denotes different types of edges, and $w$ denotes weight of the edges. Skills indicate the ability a researcher got when he/she took part in some real works. Skill ranking problem is to evaluate a researcher's specific skill in a specific field by ranking methods.

It is unconvincing to consider only the pairwise relationship between the researcher and the skill in skill ranking problem. It would be more convincing to take account of three factors. These three factors can be integrated into three kinds of relationships, including relationships between researchers and skills, researchers and fields, fields and skills. In this paper, we use the hypergraph to represent a high order relationship, which is a generation of simple network. A hypergraph can be used to represent the relationships and interactions of two or more nodes. For example, in the citation network, a paper and its reference papers can compose a hyperedge. In music recommendation problem (*Theodoridis, Kotropoulos &*

*Panagakis, 2013*; *Bu et al., 2010*), a user and the music he/she listened along with the music lists compose a hyperedge. Similar problems, such as movie recommendation, interest recommendation (*Yao et al., 2016*), news recommendation (*Liu & Dolan, 2010*; *Li & Li, 2013*), image retrieval (*Huang et al., 2010*; *Yu, Tao & Wang, 2012*), user rating (*Suo et al., 2015*), scientific ranking (*Liang & Jiang, 2016*), can be solved based on the hypergraph framework. Experiments show hypergraph methods are powerful and efficient to model multi-relationship systems. In a hypergraph, an edge, called hyperedge, contains arbitrary number of nodes, rather than pair of nodes in ordinary graphs (*Huang et al., 2010*). A hypergraph is a group of $H = (X, E)$ where $X$ is a set of vertices and hyperedge $E$ is a set of non-empty subsets of $X$. In our study, node set $X$ is consisted of researcher set $R$, field set $F$ and skill set $S$. Each hyperedge includes a researcher, a field and a skill, denoting the ternary relation among them.

## SRModel

Our SRModel aims at ranking individual's different skills using hypergraph model. We consider three kinds of objects and their relations in the model. The objects include researchers, fields and skills, forming the vertex set of the heterogeneous network. In this network, there are three-ary relations between skills, researchers and fields so that normal network structure cannot provide effective representation of this system. In this paper, we use a hypergraph to model the three-ary relations.

To construct the hypergraph model, we build a weighted heterogeneous network to calculate the weights of edges first, denoted as $H = (V, E, w)$, where $V = R \cup F \cup S$ denotes the set of vertices and $E = \{e|(v^i, v^j), i \neq j\}$ denotes the edge set. A simple example of a part of our heterogeneous network is showed in Fig. 2A. Vertex set includes three kinds of vertices, i.e., researcher $v^r$, field $v^f$ and skill $v^s$. Edge set $E$ includes three kinds of edges, i.e., edges between researchers and fields, researchers and skills, fields and skills. To understand the network clearly, the heterogeneous network can be regarded as a "tripartite network", whose vertices can be divided into three independent and unconnected node sets $R$, $F$ and $S$. And each edge $e(i, j)$ in network connects vertices $i$ and $j$ in different node sets.

There are three types of edges indicating pairwise relationship between different types of nodes. To quantify the importance of the pairwise relationship, we set each kind of edge a weight to express the importance of a node to the other, which is calculated by the ranking of node's attribute among the others in the same set.

One of the three types of edges is $e(v_j^s, v_i^f)$ between field $i$ and skill $j$. The weight between a skill and a field is calculated by the percentile ranking of the skill in the field:

$$W_1(v_j^s, v_i^f) = 1 - \gamma_{ji}^{sf} / L_i^{sf}, \qquad (1)$$

where $\gamma_{ji}^{sf}$ denotes the ranking of the skill $j$ that used in field $i$, which is calculated by ranking skills according to the times they are used in a field. $L_i^{sf}$ is the number of skills that used in the field $i$. We use the percentile ranking as the normalization to eliminate the influence of different number of skills in different fields. Besides, we subtract 1 from $\gamma_{ji}^{sf}$ to make the minimum weight not equal to zero. For example, suppose there are four skills in

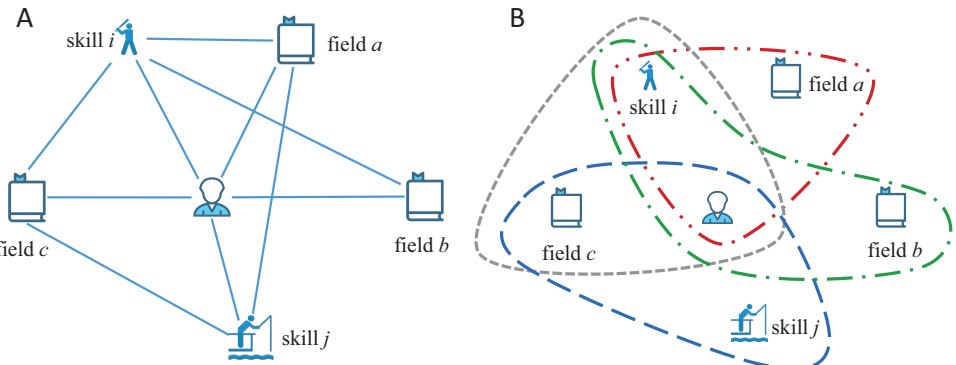

**Figure 2** **Example of our model.** (A) A simple example of a heterogeneous network containing six vertices and 10 hybrid edges. (B) An example of our SRModel, which is built based on the hyperedges. For the relationships between researcher, field and skill, a hyperedge exists only if the researcher process the skill in this field.

a field and their ranks are 1, 2, 3 and 4. The weights between those four skills and this field are 1.0, 0.75, 0.5 and 0.25, respectively.

Weight of edge $e(v_k^r, v_i^f)$ between researcher $k$ and field $i$ can be regarded as the importance of the researcher in the field, which is calculated by:

$$W_2(v_k^r, v_i^f) = 1 - \sqrt{(\gamma_{ki}^{rf} - 1)/L_i^{rf}}, \qquad (2)$$

where $\gamma_{ki}^{rf}$ denotes the ranking of researcher $k$ in field $i$, which is calculated by ranking researchers according to the numbers of previous works that they have participated in a field. $L_i^{rf}$ is the total number of researchers in field $i$. Researchers get experience and knowledge of a field when they take part in works in this field. The more works they have done, the more they learned. We perform normalization by using the percentile ranking to eliminate the influence of different number of researchers in different fields, like in Eq. (1). There are differences between Eqs. (2) and (1) because we use the square root to re-scale the weight in Eq. (2). The re-scaling operation is to make the distribution of the weight wider because there are many researchers rank in the tail (a large number of beginners). For example, suppose a field $m$ has five experts, and they have published 5, 3, 3, 1, 1 papers in this field respectively. Thus, the weights between those five experts and field $m$ are 1.0, 0.553, 0.553, 0.225, 0.225, respectively.

Similarly, weight of edge $e(v_k^r, v_j^s)$ between researcher $k$ and skill $j$ represents how important a researcher is in a given skill, computed by:

$$W_3(v_k^r, v_j^s) = 1 - \sqrt{(\gamma_{kj}^{rs} - 1)/L_j^{rs}}, \qquad (3)$$

where $\gamma_{kj}^{rs}$ denotes the ranking of researcher $k$ with skill $j$, which is calculated by ranking researchers according to the times they used this skill in their previous works. $L_j^{rs}$ denotes the number of researchers with skill $j$.

A hypergraph $H^h = (X, E^h, w^h)$ is built after constructing the heterogeneous network and calculating the weights of edges. A hypergraph built on the heterogeneous network

in Fig. 2A is demonstrated in Fig. 2B. In $H^h$, $X$ is a set of hybrid nodes, composed by researcher, skill and field. $E^h$ is hyperedge on $X$, and $w^h$ is a weight function of hyperedge. For the relationships between researcher, field and skill, a hyperedge exists only if the researcher processes the skill in this field. We define the weight function of hyperedge inspired by the Laplace kernel function, and the distance calculation method in (*Huang et al., 2010*). The weight function is:

$$W(v_k^r, v_i^f, v_j^s) = e^\Theta, and \tag{4}$$

$$\Theta = \frac{W_1(v_j^s, v_i^f)}{2 \cdot \overline{W_1}} + \frac{W_2(v_k^r, v_i^f)}{2 \cdot \overline{W_2}} + \frac{W_3(v_k^r, v_j^s)}{2 \cdot \overline{W_3}}, \tag{5}$$

where $\overline{W_1}, \overline{W_2}$ and $\overline{W_3}$ are the average edge weights between fields and skills, researchers and fields, researchers and skills, respectively.

Based on the above definition and introduction, our SRModel has been formulated. The value of skill ranking of a researcher is calculated by the weight of the hyperedge in the hypergraph.

# EXPERIMENT SETTING

In this section, we introduce the dataset we used in our experiment. Then, we construct the hypergraph and get the skill sets as well as the skill ranking of researchers.

## Dataset construction

Previous studies on researcher evaluation usually use datasets of publications to evaluate their methods. These datasets include Digital Bibliography & Library Project (DBLP), American Physical Society (APS), Microsoft Academic Graph (MAG). However, authors' skills are unavailable in those prevalent datasets. In the previous work where skills are required, such as team formation problem, team member replacement, collaborators recommendation, researchers usually use terms extracted from keywords, title (*Farhadi et al., 2011*) or conference and journal name (*Li et al., 2017*) instead of authors' real skills. Either terms in titles or journal names have limitations to reflect skills of researchers, because the division of the work is not clear. However, there are several journals providing authors' contribution statement, such as the British Medical Journal, Nature, Lancet. Besides, journals such as the FEBS Journal and PLOS journals require authors to declare their contributions in a specified format while submitting the paper.

We create a dataset by crawling papers' information together with their research fields and authors' individual contributions from the PLOS ONE journal website to validate our model, as we mentioned in the Background section. PLOS ONE journal is an open-access journal, so we can extract information from their website by parsing their HTML pages of each paper with Python. The contribution information of this journal has been used to analyze the authors' contribution pattern recently (*Corrêa Jr. et al., 2017*; *Sauermann & Haeussler, 2017*). The details of our dataset are shown in Table 1. At first, we remove papers with incomplete information. In the raw dataset, 28,748 kinds of contributions are found, and most of them have similar meanings. PLOS journals recently adopted the CRediT

**Table 1  Selected PLOS ONE dataset.**

| Items | Raw | Preprocessed |
|---|---|---|
| Time Span | 2006-2017.07 | 2006-2017.07 |
| # field | 11 | 11 |
| # paper | 182,007 | **164,543** |
| # author | 676,852 | **684,844** |
| # contribution | 28,748 | **10** |

Taxonomy providing standardized and fine-grained information of authors' contribution (*Atkins, 2016*), but there are no standard rules for naming paper's contributions in the early years. Using the raw contribution directly may bring inaccuracy. There are 100 items appearing more than 40 times accounting for the vast majority (about 96%). Thus, we cluster 100 kinds of contributions that appear frequently into 10 categories manually. The categories are shown in Table 2. Here, we regard these 10 categories of contributions as skills of researchers.

The acronyms of authors' name followed the contributions on the PLOS ONE web pages. We match the acronyms up with authors' full name in author list of each paper. Then we get the authors' skills in every paper.

Another vital step is name disambiguation. Multiple authors may share the same name, and there is no unique identifier for each author in PLOS ONE journal. It can cause confusion and even fault if we regard those authors with the same name as one person. Many works have employed a variety of features to distinguish authors sharing the same name, such as affiliations, coauthors, topics or research interests (*Kang et al., 2009*; *Ferreira, Gonçalves & Laender, 2012*). To distinguish authors in PLOS ONE dataset, the following features are adopted: (i) Affiliation: authors with the same name and the same affiliation are regarded as the same authors; (ii) Coauthor: if two authors $M_1$ and $M_2$ with the same name coauthored one or more papers with a third author $N$, it is likely that $M_1$ and $M_2$ are the same author named $M$. We combine these two features to disambiguate the authors' name. There are 676,852 authors in the raw dataset, and 684,844 authors after naming disambiguation.

## Network construction

To construct the SRModel framework using PLOS ONE dataset, our experiment consists of four main parts as follows.

1. Construct a heterogeneous network with three kinds of vertices and three kinds of edges.
2. Compute edge weights according to Eqs(1)–(3).
3. Build a hypergraph $H^h = (X, E^h, w^h)$ and compute hyperedge weights using Eq. (4).
4. Calculate the researchers' yearly skill ranking $W^i$ from 2006 to 2016 respectively.

In the first step, we construct a heterogeneous network including three kinds of vertices and three kinds of relationships. Edge between researcher and field/skill exists only if he/she published paper in this field or used this skill at least once. A vertex denoting field $v_i^f$ and a vertex denoting skill $v_j^s$ are connected if one or more papers in field $i$ use skill $j$. There are

**Table 2  The 10 skills and their composed contributions.**

| Skills | Contributions in PLOS ONE dataset |
| --- | --- |
| Data analysis | Analyzed the data, Data curation, Statistical analysis, Interpretation of data, Interpreted the data, Data interpretation, Interpreted data, Analysis and interpretation of data, Performed statistical analysis, Interpretation of the data, Performed the statistical analysis |
| Study and experiments design | Conceived and designed the experiments, Designed the study, Conceived and designed the study, Designed the experiments, Conceived the study |
| Experiment performing | Performed the experiments,Visualization, Performed the experiments |
| Paper writing | Wrote the paper, Writing original draft, Wrote the manuscript, Contributed to the writing of the manuscript, Writing - original draft, Drafted the manuscript, Wrote the first draft of the manuscript |
| Software and tools | Contributed reagents/materials/analysis tools, Software, Designed the software used in analysis, Contributed reagents/materials/analysis tools |
| Theory analysis | Methodology, Conceptualization, Formal analysis, Validation, Interpreted the results, Interpretation of results |
| Paper reviewing | Writing review & editing, Revised the manuscript, Writing - review & editing, Edited the manuscript, Read and approved the final manuscript, Reviewed the manuscript, Critically revised the manuscript, Final approval of the version to be published, etc. 44 items in total |
| Data collection | Investigation, Resources, Acquisition of data, Data collection, Collected the data, Collected data, Obtained permission for use of cell line, Sample collection, Data acquisition, Data acquisition, Collected samples, Collected the samples |
| Supervision | Supervision, Project administration, Study supervision, Supervised the study, Supervised the project, Supervised the work |
| Funding acquisition | Funding acquisition, Obtained funding, Financial support |

684,844 authors, 11 fields and 10 skills in the heterogeneous network, forming 2,758,522 edges in total.

In the second step, we compute the weights of edges in the constructed network. We compute the weights of field-skill, researcher-field and researcher-skill by Eqs. (1)–(3), respectively. We rank the skills in field $i$ by the numbers of papers in field $i$ that using the skills, and for each skill $j$ the ranking is denoted as $\gamma_{ji}^{sf}$. $L_i^{sf}$ is the total number skills in field $i$. When calculating the weights between researchers and fields, we assume the more paper a researcher published in a field, the more he/she is familiar with the field. Thus, we rank the researchers according to the numbers of papers they published in field $i$, and for researcher $k$ the ranking is denoted as $\gamma_{ki}^{rf}$. Let the total number of researchers in field $i$ as $L_i^{rf}$ in Eq. (2). Similarly, $\gamma_{kj}^{rs}$ is the ranking of researcher $k$ using skill $j$ by the number of

---

**Algorithm 1:** Algorithm to calculate the skill ranking of researchers

**Input:** Weight of edges; paper information
**Output:** Hyperedge weights

1   $\overline{W}_1 = AVG(skills, fields)$;
2   $\overline{W}_2 = AVG(researchers, fields)$;
3   $\overline{W}_3 = AVG(researchers, skills)$;
4   $hyperedges = list()$;
5   $hyperWeight = dict()$;
6   **for** *paper in dataset* **do**
7      add (researcher, field, skill) to *hyperedges*;

8   **for** $(r, f, s)$ *in hyperedges* **do**
9      $\Theta = \frac{w_1[s,f]}{2 \cdot W_1} + \frac{w_2[r,f]}{2 \cdot W_2} + \frac{w_3[r,s]}{2 \cdot W_3}$;
10      $weight = e^{\Theta}$;
11      $hyperWeight[r, f, s] = weight$;

12   return *hyperWeight*;

---

papers he published using this skill, and $L_j^{rs}$ is the total number of researchers that using skill $j$ in Eq. (3).

In the third step, we construct the hypergraph and calculate the skill proficiency of authors. A hyperedge connects a researcher, field and skill if the researcher published papers in this field using this skill. There are 9,501,866 hyperedges in the constructed hypergraph in total. And the weight of hyperedge is calculated by Eq. (4), representing a researcher's skill proficiency in a field. The pseudo code of the algorithm to calculate the skill ranking is shown in Algorithm 1. Let $E$ denote the number of hyperedges, so the time complexity of constructing a heterogeneous network of skill, field and researcher is $O(E)$. Then, building hypergraph based on the network and calculating weights of hyperedges can be performed together. The time complexity of these two steps is also $O(E)$. So the overall time complexity of our skill ranking method is $O(E)$.

In fact, the skill ranking that we calculate in the third step is a snapshot at the time when the data are collected. In the last step, we take a snapshot every year to construct a series of hypergraphs and calculate the weights of hyperedges to represent the researchers' skill rankings at different time, denoted as $W^i$, where $i \in [2006, 2016]$. That is, the skill ranking of the researcher in a field at year $i$ is calculated by all the papers he published until year $i$. We use the yearly skill ranking information to analyze the dynamic of researchers' skill proficiency. Also, we can get the researchers' abilities in the year when the paper was published.

# RESULT

We construct the hypergraph model and then calculate the weight of the hyperedge using it. Based on the hyperedge weight, we can calculate the skill ranking of a researcher in a

**Table 3  Correlation coefficient of skill ranking and h-index**

| Skill | Correlation coefficient |
| --- | --- |
| Experiment performing | 0.7717 |
| Data analysis | 0.8036 |
| Paper reviewing | 0.7013 |
| Paper writing | 0.801 |
| Supervision | 0.5561 |
| Theory analysis | 0.5177 |
| Funding acquisition | 0.7582 |
| Software and tools | 0.7822 |
| Study and experiments design | 0.8155 |
| Data collection | 0.6173 |

field. In this section, we validate the effectiveness of the SRModel and carry out several analyses on the ranking results.

## Validation of SRModel

We computed the correlation coefficient between researchers' skill ranking and their h-index and performed a hypothesis test to validate our model's performance.

### Correlation between skill ranking and h-index

To verify the validity of the method, and explore the potential relationship between skill ranking and h-index, we analyze how skill ranking changes with the increase of h-index. PLOS One is a journal mainly focusing on biology. Among the 182,007 papers that we obtained in their website, more than 169,150 papers are in the field of biology. Thus, we use the papers and authors in biology to perform the verification. Researchers' skill rankings are calculated by the data in PLOS One. But only a fraction of researchers' papers is published in this journal. It is not reasonable to use authors' real h-indices to analyze the relationship with their skill rankings. Thus, we calculate the h-indices of all the authors in PLOS ONE dataset according to the papers they published in this journal and their citation counts. The h-indices of authors in PLOS One dataset range from 0 to 32. Pearson's correlation coefficient quantifies the linear correlation between two groups of continuous variables (*Pearson, 1895*). Thus, we calculate the correlation coefficient between skill ranking and h-index, shown in Table 3. We notice that all the correlation coefficients are larger than 0.5, especially for the skill "Data analysis", "Paper writing" and "Study and experiments design", of which the correlation coefficients are larger than 0.8, which indicates a high correlation. This result means that a researcher with a higher h-index often has a higher skill ranking, which suggests that if a researcher is more skillful with academic skills, they can obtain outcomes with higher quality.

We then analyze the distribution of skill ranking to prove the rationality of the model. The result is shown in Fig. 3. Both of two distribution curves subject to exponential distribution of function $y = e^{a+bx+cx^2}$, and the R-square of these two fitting functions is close to 1, which indicates the fitting degree is high. It means that researchers' h-index and

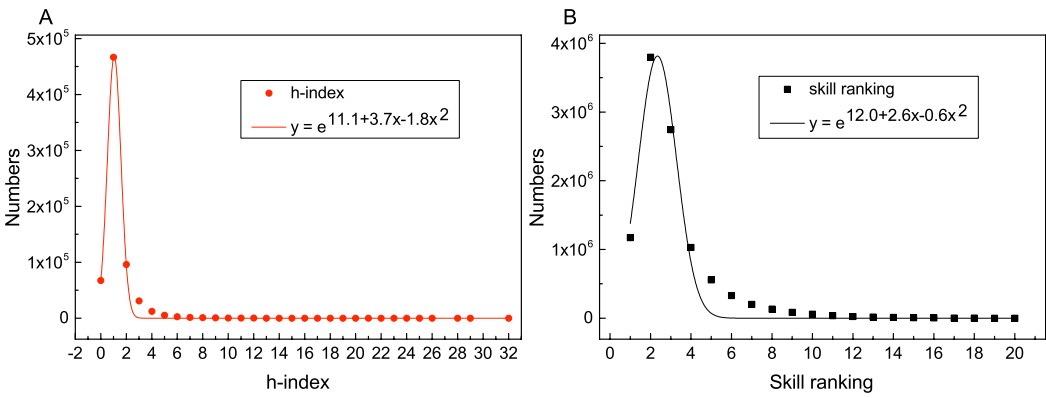

**Figure 3** Distributions and fitted lines of h-index ($R^2 = 0.99$) and skill ranking ($R^2 = 0.97$). The $x$ axes denote the value of h-index and skill ranking, and $y$ axes denote the numbers. Both h-index and skill ranking are subject to exponential distribution.

skill ranking have the same kind of distribution, which also explains the high correlation coefficient between them and the rationality of the SRModel.

### Relationship between skill ranking and citation

To validate whether the skill ranking computed by our model is significant for the quality of scholar's research outputs, we define a hypothesis that there are differences between the citation counts of papers whose authors' average skill rankings are different. We use the single side $t$-test, which is also known as "student test" (*Gosset, 1908*), to test the hypothesis we made. In our experiments, we use papers in biology from 2009 to 2015 as the sample set. Here, firstly we group all papers by their publish years because the citation is highly influenced by paper's age and we just have the citation counts of paper in We remove the first three years because we don't have enough papers and researchers in our dataset, and we remove the last two year because the citation counts of these papers are still growing. Then, we calculate the average skill ranking of paper's authors in the year they wrote it for each paper. We split every group into two parts to perform our test. The one consists of papers with authors average skill ranking less than the median value of the group. The other contains the rest of papers. We denote the part of papers with higher authors' skill ranking as random variable $X_1$ and the lower papers as $X_2$.

We perform a $t$-test with the settings mentioned before. The *null hypothesis* for testing is $H_0 : \mu_1 = \mu_2$ and *alternative hypothesis* as $H_1 : \mu_1 > \mu_2$, where $\mu_1$ and $\mu_2$ are the mean values of the population of the two groups of random variables we defined above. We use the SciPy module in Python to do the test (*Jones, Oliphant & Peterson, 2001*), which can compute both the statistical value and $p$-value of the $t$-test together by function

$$scipy.stats.ttest\_ind(X_1, X_2, equal\_var = True).$$

If the $p$-value is less than a significance level (typically 0.05), we can reject the null hypothesis and hold the alternative hypothesis $H_1 : \mu_1 > \mu_2$, which means the average citation of papers with higher skill ranking authors are more than papers with authors

**Table 4** Average citation counts of two groups of samples and their *p*-value.

| Year | Avg-low | Avg-high | *p*-value |
|------|---------|----------|-----------|
| 2009 | 37.5 | 42.0 | $7.2 \times 10^{-4}$ |
| 2010 | 31.2 | 36.0 | $7.0 \times 10^{-5}$ |
| 2011 | 24.8 | 27.5 | $2.3 \times 10^{-6}$ |
| 2012 | 18.5 | 20.4 | $1.6 \times 10^{-11}$ |
| 2013 | 13.3 | 14.7 | $1.9 \times 10^{-14}$ |
| 2014 | 9.0 | 9.9 | $1.2 \times 10^{-13}$ |
| 2015 | 4.8 | 5.3 | $7.7 \times 10^{-13}$ |

lower skill ranking. But the result of *t*-test is sensitive to the equality of samples variance. If variances of two groups of samples are not equal, the result needs to be corrected. So before performing the *t*-test on samples, we do a Levene's test to assess the equality of two groups of variances. If the two variables are not equal, we need to set the parameter *equal_val = False* in the *t*-test function to avoid deviation. The average citation counts of two groups of samples are shown in Table 4. Papers written by authors with relatively low skill ranking have lower average citation counts. All the *p*-values in *t*-test are lower than 0.05, which indicates there are significant differences in sample mean values. Thus, the skill ranking of authors influence the citation counts of outputs.

## Analysis of skill ranking increase

We define the researcher's academic age in PLOS ONE dataset as the number of years between his first paper and the last paper in this dataset. Although PLOS ONE is a comprehensive journal with a large number of papers, it cannot include all the papers published by every researcher. There still exist many researchers that published many papers but only one or two in this journal. Analyzing the skill rankings of those researchers is meaningless. In order to avoid the defect of the dataset as much as possible, we choose researchers whose academic ages and numbers of published papers are both in the top 10% respectively, named as "top researchers", to carry out the analysis. In the PLOS ONE dataset, there are 26,978 "top researcher".

We first calculate the yearly change rate of researchers' skill ranking. The yearly change rate indicates that how much the skill ranking of a researcher increased or decreased compared to that in one year before, defined as $\Delta_i = \frac{W_i - W_{i-1}}{W_{i-1}}$, where $W_i$ is the value of skill ranking in year $i$ and $W_{i-1} \neq 0$. We show the distribution of yearly change rate between 0 to 100% in Fig. 4. The change rate of most researchers' skills is less than 20%. The green bar demonstrates a slight decrease existing in the researchers' skill ranking, because the progress of other researchers decreases the rankings of them. Observing the curve of researchers' skills, we found the skills of most researchers are fluctuating upward, but the rising patterns are different. Then, we explore the commonalities and differences of the changes for researchers' different skills.

First, we analyze the rising stages of skills. When a researcher's skill ranking increased more than 20 % in a year, we call this year a "fast increase year" for his skill. The distribution of the longest continuous rapid growth years of each set of researcher-field-skill is presented

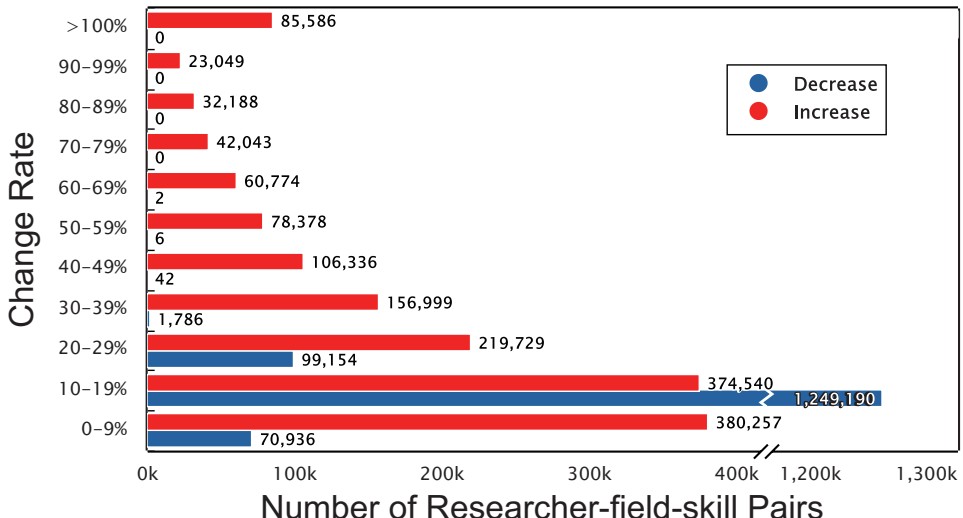

**Figure 4** **Yearly change rate of top researchers' skills.** The *x*-axis is the number of researcher-field-skill pairs and the *y*-axis is the change rate. Red and blue bars denote the number of increase and decrease sets with the corresponding change rate in *y*-axis, respectively.

in Fig. 5. In Fig. 5, we count the total number of researcher-field-skill sets for different length of time. About 434 thousand researcher-field-skill sets experience one-year rapid growth, and the number of that for two years reduced to around 220 thousand. A few skill sets rise continuously more than five years, including 1,254 sets for 5 years and 53 sets for 6 years. The average fast-growing years among the 730,789 researcher-field-skill sets for 26,798 researchers is 1.5 years. That is, it is generally for researchers to spend one or two years putting their energy on one skill to have a good command of it, and then they may focus on other disciplines or other skills. Several reasons account for this phenomenon. First, researchers usually change their research fields in their research careers, especially for those interdisciplinary researchers. The transformation of fields can bring researchers more inspiration and creativity. Second, for a researcher, as he becomes more and more skillful, the work he undertakes in the team may change. We then explore the skill transfer of researchers.

## Changing patterns of skill ranking

Then, we explore the changing patterns of different skills. To investigate how skill ranking changes over time, we propose a method to count the average ranking for each skill in different periods. There are three problems needing to be considered. (1) Researchers' academic ages are different, and the time when they enroll in a skill or field varies. (2) Many researchers didn't publish paper on PLOS ONE at the beginning of their academic career. Studying the skill transfer of them is unconvincing. (3) Many researchers didn't publish papers every year in PLOS ONE and the numbers of paper published by researchers are totally different. To solve these problems, we define a researcher's "academic stage" as the period between years he has published papers. For example, a researcher *m* published two, five and one papers in 2013, 2015 and 2016, respectively, so he has three academic

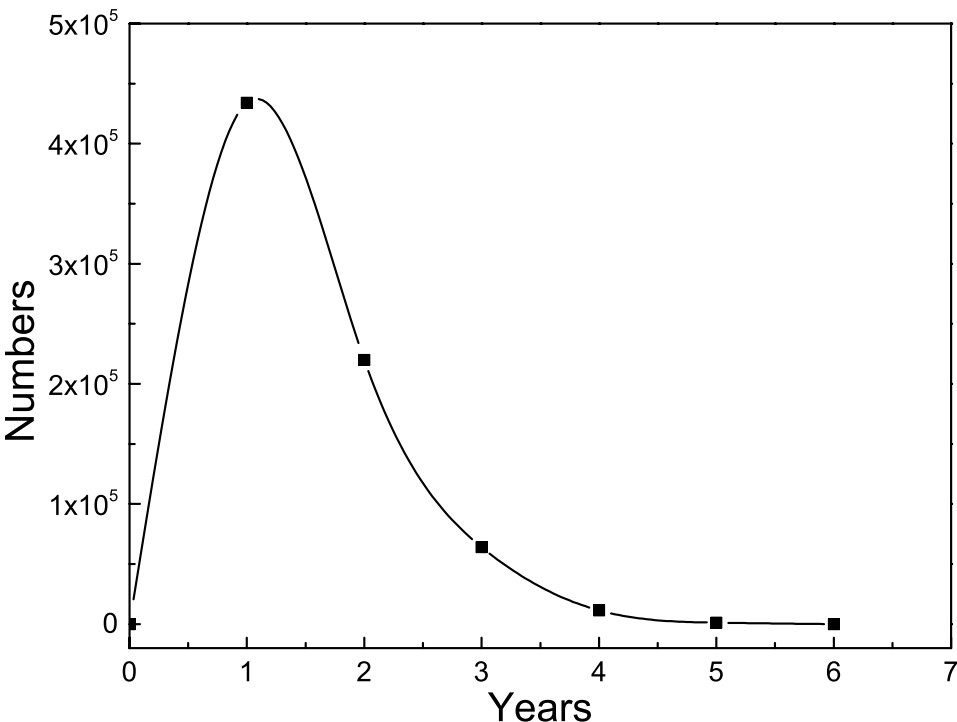

**Figure 5** **Continuous rapid growth years of researchers' skills.** The $x$-axis denotes the continuous rapid growth years and $y$-axis denotes the number of researchers' skills. There are 730,789 pairs of (researcher, field, skill) for the 26,978 top researchers in total.

stages, which are 2013–2014, 2015–2015, 2016–2016. The skill ranking of each stage is indicated by the skill ranking in the first year of the stage.

After we got the skill ranking of all the stages for all the top researchers, we calculate the average value in each stage for different skills, and show the result in Fig. 6. We divide the skills into four groups according to their trend. In Fig. 6A we notice that there are five skills of which the ranking keep increasing throughout all academic stage, and the increase rate is higher than other groups, including "paper writing", "data analysis", "study and experiments design", "experiment performing" and "software and tools". It suggests that these five skills are the basic academic skills so that they are important for many researchers throughout their academic careers. The second group, as shown in Fig. 6B, has two skills, "supervision" and "theory analysis". The rankings of these two skills change slowly in the early stages and have a sharp increase at the latter academic stages, which indicates that these two skills need more experience in academia and are harder to develop. Figure 6C is the third group of skills, including "paper reviewing" and "data collection". These two skills' ranking rise in the early stage and finally falling slightly, especially the skill "data collection", whose increase rate is very small after the fourth stage. We assume that these two skills are easy to get started, and when researchers have more experience they will not use these skills frequently. There is one skill in Fig. 6D, the trend of which is not so obvious.

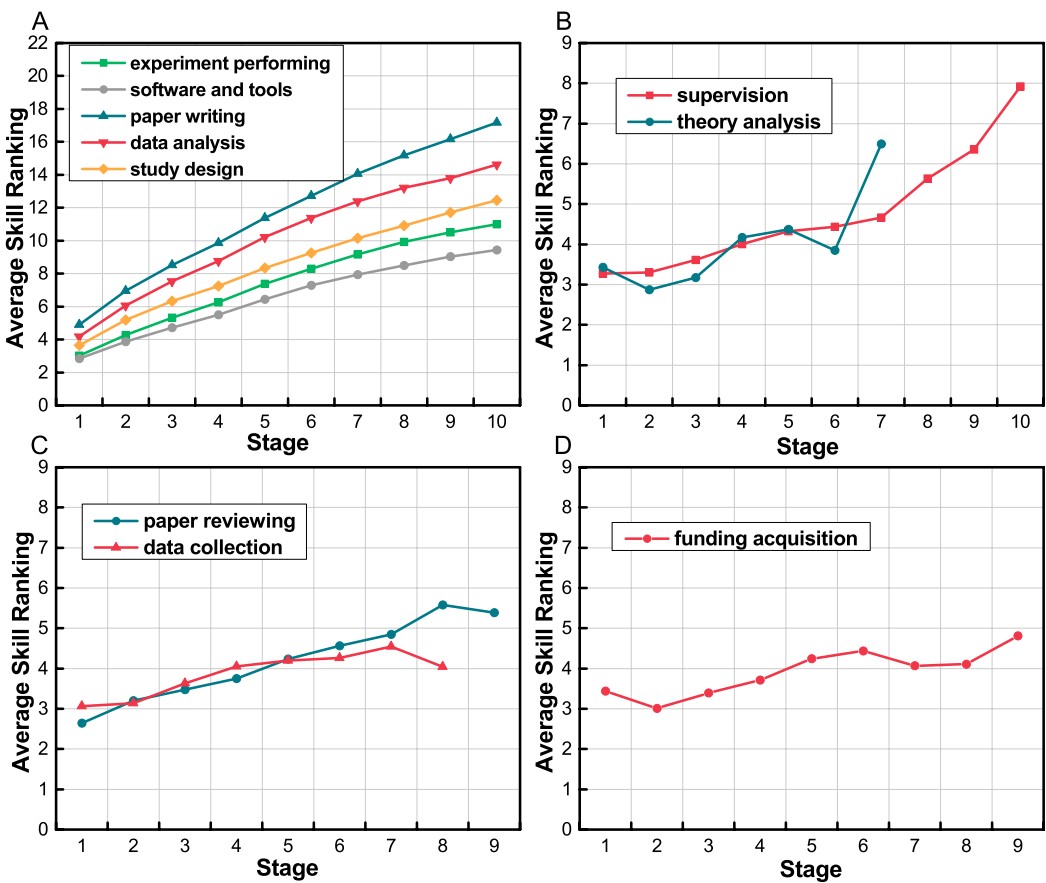

**Figure 6** **Average skill ranking in different periods.** The *x*-axis is the stage, and *y*-axis is the researchers' average skill ranking in each stage. We divide the skills into four groups according to their trends. (A) Basic skills that increase throughout researchers' academic careers. (B) Skills that need more experience to be developed. (C) Skills that are easy to get started. (D) A skill whose trend is not so obvious.

We think "funding acquisition" has little correlation with time in our dataset. Maybe the time span is not long enough to find its changing law.

Thus, we find the changing patterns of different skills vary. Some skills are easy to get started. When researchers have more experience, they will transfer their interests to other skills. Some skills are harder to develop, researchers with these skills need to develop other skills first. Thus, they develop these skills in their later academic stage. There are also some basic skills are important for many researchers throughout their academic careers.

## CONCLUSION AND FUTURE WORK

In this paper, we make both empirical analysis and data analysis to figure out factors affecting the result of skill ranking. Then, we construct a model named SRModel based on hypergraph to describe the relationships among researchers, skills and fields. This model can be used to rank a researcher's scientific skills. We applied our model on the PLOS ONE dataset, which is extracted from the journal website. We obtained the weighted field-skill

sets for each researcher. We validated our method by correlation analysis with h-index and hypothesis test. Then, we use the results to analyze the increase of skill ranking and patterns of skill ranking change.

Skill ranking can be applied to many practical problems, such as collaborator recommendation, team formation, team member replacement and team refinement. In problems where skill similarity is required, the researcher's proficiency in each skill can make the results more precise. In other problems where skill grading is needed, a fine-grained method can lead to a better result. In future work, we will use our model to solve some practical problems and examine its reliability and effectiveness. Datasets of other disciplines like software engineering and physics can be taken into consideration to verify the validity of the model. Besides, more factors and relationships will be taken into consideration to rank skills.

### Funding

This work was supported by the Fund for Promoting the Reform of Higher Education by Using Big Data Technology, Energizing Teachers and Students to Explore the Future (2017A01002), the Fundamental Research Funds for the Central Universities (DUT18JC09), Liaoning Provincial Key R&D Guidance Project (2018104021), and the Liaoning Provincial Natural Fund Guidance Plan (20180550011). There was no additional external funding received for this study. The funders had no role in study design, data collection and analysis, decision to publish, or preparation of the manuscript.

### Grant Disclosures

The following grant information was disclosed by the authors:
Using Big Data Technology, Energizing Teachers and Students to Explore the Future: 2017A01002.
Fundamental Research Funds for the Central Universities: DUT18JC09.
Liaoning Provincial Key R&D Guidance Project: 2018104021.
Liaoning Provincial Natural Fund Guidance Plan: 20180550011.

### Competing Interests

Xiangjie Kong is an Academic Editor for PeerJ.

### Author Contributions

- Xiangjie Kong conceived and designed the experiments, analyzed the data, contributed reagents/materials/analysis tools, prepared figures and/or tables, performed the computation work, authored or reviewed drafts of the paper, approved the final draft.
- Lei Liu conceived and designed the experiments, performed the experiments, analyzed the data, contributed reagents/materials/analysis tools, prepared figures and/or tables, performed the computation work, authored or reviewed drafts of the paper, approved the final draft.

- Shuo Yu performed the experiments, analyzed the data, authored or reviewed drafts of the paper, approved the final draft.
- Andong Yang analyzed the data, authored or reviewed drafts of the paper, approved the final draft.
- Xiaomei Bai and Bo Xu conceived and designed the experiments, analyzed the data, authored or reviewed drafts of the paper, approved the final draft.

## Data Availability

The raw code are available in the Supplemental File.

## Supplemental Information

Supplemental information for this article can be found online at http://dx.doi.org/10.7717/peerj-cs.182#supplemental-information.

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
