# Peer review of "Skill ranking of researchers via hypergraph"

_PeerJ Computer Science, doi:10.7717/peerj-cs.182_

## Round 0.1 · original submission · Major Revisions

Dear X. Bai,

In addition to the reviewers' comments, please take special attention to related literature on patterns of authors contributions in scientific literature. We look forward to receiving a revised version of your manuscript.

Best regards,

Reviewer 1 ·

Basic reporting

In this manuscript, the authors use a methodology for analyzing scientific skills and employ a method for evaluation. The authors represent the relationship between researchers, fields, and skills through a tripartite graph, which is converted into a hypergraph.

1 - The quality of the figures is not appropriate regarding the number of DPI. So, it is difficult to read the numbers. In particular, I believe that the graphical quality of Figure 2b shall be improved.

2 - The overall English shall be improved. Some examples are presented as follows:
- When you use the word "heterogenous", I think you meant "heterogeneous". Additionally, you used “data set” instead of “dataset”.

- In the abstract, the phrase “Finally, we analyze the pattern how...”, a preposition is missing after the word “pattern”;

-In introduction:
- instead of "Scientific cross-disciplinary collaboration brings positive effect to...". you should write "Scientific cross-disciplinary collaboration brings positive effects to...";
- "A dozen of state-of-art works have been proposed..." should be "A dozen state-of-art works has been proposed...";
- "The index has been proved to be an simple..." should be "The index has been proved to be a simple...".
- "In recent work, methods of ranking authors in heterogeneous network..." can be replaced by "In recent work, methods of ranking authors in an heterogeneous network..."
- " They can not reflect..." should be " They cannot reflect..."

In Feature description:
- “Facing the high competitive...” should be “Facing the highly competitive...”
- “… and researcher’s skill ...” should be “and the researcher’s skill ”

Note that I depicted here only some cases of grammar issues. So, there are many others, and I believe that the writing shall be better revised.

3 - b) You used the letters $\gamma$ and $W$ in equations (1), (2), (3), and (4). It can confuse the reader. In the case of equations 1, 2, and 3, Iou could use $W_1$, $W_3$, and $W_3$, respectively, as adopted in equation (5).

4 - I believe that you could better explain about the two versions of ranks ($\gamma$).

5 - In line 220, you mentioned a section called " Feature", but the section name is "Feature description".

6 - In equation 7, information regarding the variables $n_1$ and $n_2$ are missing.

7 – The measurement presented in Figure 5 is not clear.

8 - When you mentioned that "the transformation of the field brings researchers more inspiration and creativity, contributing to the success of their scientific researches", I think that you cannot make this affirmation because you do not have enought elements. To fix this problem, you can say that "the transformation of the field can bring...".

Experimental design

1- I'm not sure about the result presented in Figure 4. Because of the log scale in the y-axis, the curves seem to follow the same behavior. However, if you compare the differences in skill rankings between the points in which h-indexes are 1 and 19, the difference is in the order of magnitude of 10^4. In other words, the slopes of the curves are much different.

2- In Hypothesis test section, I could not understand why to validate your method you considered that "there are significant difference between the paper citation
counts and the average skill rank of its authors". This information is critical in this section and shall be better explained.

Validity of the findings

1 - You affirmed that
“It has been found that collaborations cross disciplines are more competitive and more likely to gain high quality outputs. First, the development of the collaborative relationships between multi-disciplines makes researchers realize the advantages of resource sharing, which can make a better data and knowledge utilization either within a team or in the whole society.”
However, you cannot say that the articles that are most frequent have more quality or are more competitive.

2 - I did not understand the following text: "We start our analysis by calculating the change rate of researchers’ skill ranking in every two years. We have calculated the skill ranking of researchers every year, as referred in Network Construction section. We calculate the change rate of skill ranking between every two adjacent year i and i − 1, that is..."
Because of this part of the text, I could not comprehend the analysis regarding Figure 5.

Additional comments

The subject and the proposed model are valuable and can be applied in other related problems. So, I believe that the article is suitable to be published in "PeerJ - Computer Science journal" after a major revision.

Reviewer 2 ·

Basic reporting

no comment

Experimental design

no comment

Validity of the findings

no comment

Additional comments

The authors present a new model on factors related to researchers' skill.
The tool considered in the model is a hypergraph (a heterogeneous network).
All experiments were carried out considering the PloS One bibliographic dataset
(more than 164000 papers coauthored by more than 684000 researchers).

In general, the treatment of the problem is technically sound. However, there
are some questions regarding the presentation and results which require
attention:

* 1. The conclusion stated in the Abstract seems to me obvious: "Our studies
show that many researchers are playing different roles in the different stages
of their academic career." Please explain in more detail the contribution of the
work.

* 2. The description and the mathematical formulation to construct the SRModel
framework models were well introduced in this work. Would it be possible to
present a summary of all the steps in the form of an algorithm? About the
arithmetic complexity of the considered steps, how are the computing time and
memory usage?

* 3. Figure 2 lacks details. Fig. 1(a): Please explain what are the 11 hybrid vertices in the graph. Fig.2(b): I did not understand (properly) how are built the hyper edges.

* 4. What was the data source considered for the h-indices? Please justify
the choice of data source, and indicate the period of data collection.

* 5. The review of prior literature is reasonable. The reference list should be carefully reviewed. For instance:
- Rahman, M. T., Mac Regenstein, J., Kassim, N. L. A., and Haque, N. (2017). The need to quantify
authorsŕelative intellectual contributions in a multi-author paper. Journal of Informetrics, 11(1):275-81

Reviewer 3 ·

Basic reporting

The manuscript entitled "Skill Ranking of Researchers via Hypergraph" presents a study about the ranking of researchers using hypergraph. I have the following comments

- I strongly recommend that the manuscript be reviewed by an expert in the English language for both the composition of the sentences and the grammar review. There are many wrong expressions in many parts of the text. Although the main topic is very interesting to the readers of this journal, the lack of text correctedness affects the quality of the manuscript.

- The related works are well structured and the main background of the manuscript is provided. The raw data has also been provided by the authors.

- The caption of Figure 1 is wrong. The correct form could be "The distribution of number of fields for ALL the papers".

- The drawing of Figure 2-b) needs to be improved.

- The caption of Figure 3 is confusing. I would remove this figure because the information that is shown in the pie chart is clearly discussed in the text and doesn't need illustration.

- The sections of the manuscript are well structured, however the name of the second section ("Feature description") is very generic and does not reflect its content. This section looks more like a background/conceptualization section.

- The section "Analysis Methods" (which includes the definitions of Correlation Coefficient and Hypothesis Test) is not necessary because it brings definitions that are very well known.

- The name of the subsection "Hypothesis test" should be changed for something that describes the purpose of the section.

Experimental design

The main topic of the manuscript is within the scope of this journal and the research questions are well defined.

- In the section "Dataset Construction" the authors mention that there are no standard rules for naming contributions in PLoS One journal. However, nowadays the journal adopts the CRediT Taxonomy to describe the author's individual contributions. This should be clarified in the text.

- It is not clear in the section "SRModel" how the ranks are defined in equations 1, 2 and 3, which are then used to define the weight of the edges.

Validity of the findings

The authors present an interesting method which performs skill ranking analysis of researchers. I think the main contribution is the analysis of the changing pattern of skill ranking. But I would like to highlight the following issues with the proposed method.

What are the challenges of using data from other journals? As the authors wrote, PLoS One cannot include all the papers published by each researcher. Using data only from this journal can be a source of bias. Can the same procedure be employed to construct a similar dataset from other journals? What would be the limitations of such approach?

Since the distributions of skill ranking and h-index are very similar and they present a high correlation, what is the novelty of the proposed approach concerning its applicability? A discussion on what are practical problems on which the skill raking can be applied is still missing in text.

One of the contributions listed by the authors is "Data retrieving". What do they mean in this case? The creation of a dataset? In this case, it should be detailed in the text how this dataset can be accessed and manipulated by the readers.

---

## Round 0.2 · Minor Revisions

Improvements have been observed from the previous version, however, one referee has pointed out that the text should be improved regarding style and language.

Reviewer 1 ·

Basic reporting

The majority of the pointed issues were fixed.

Experimental design

no comment

Validity of the findings

no comment

Additional comments

1 - I believe that the writing shall be improved. There are still many grammar mistakes.

2 - Regarding the figures, you could consider using a vectorial format (e.g., pdf or eps).

3 - In the case of Fig. 6, you could include the axis labels.

Reviewer 3 ·

Basic reporting

no comments

Experimental design

no comments

Validity of the findings

no comments

Additional comments

All the reviewers' suggestions were addressed and the manuscript was improved for what concerns the clarity of the presented methodology and results, as well as, the text correctedness.

---

## Round 0.3 · accepted · Accept

Please provide all figures in a vector format, as pointed out by one of the referees.

Reviewer 1 ·

Basic reporting

The text is much better than in the previous version. Besides, almost all the reviewers' suggestions were addressed. So, the manuscript is suitable to be published. The single remark is that the images are still not in vector image format because when you zoom the images it is possible to see the pixels (the single figure that is in vector format is Fig. 5).

Experimental design

no comment

Validity of the findings

no comment

Additional comments

no comment

Reviewer 2 ·

Basic reporting

no comment

Experimental design

no comment

Validity of the findings

no comment

Additional comments

The paper has been revised accordingly.

The authors thoroughly clarified all the points I raised in my previous review. I have no further objections against publication.